# CD117, BAP1, MTAP, and TdT Is a Useful Immunohistochemical Panel to Distinguish Thymoma from Thymic Carcinoma

**DOI:** 10.3390/cancers14092299

**Published:** 2022-05-05

**Authors:** Mounika Angirekula, Sindy Y Chang, Sarah M. Jenkins, Patricia T. Greipp, William R. Sukov, Randolph S. Marks, Kenneth R. Olivier, Stephen D. Cassivi, Anja C Roden

**Affiliations:** 1Department of Laboratory Medicine and Pathology, Mayo Clinic, Rochester, MN 55902, USA; mounika.angirekula@ascension.org (M.A.); singyun.chang@fraserhealth.ca (S.Y.C.); greipp.patricia@mayo.edu (P.T.G.); sukov.william@mayo.edu (W.R.S.); 2Department of Quantitative Health Sciences, Mayo Clinic, Rochester, MN 55902, USA; jenkins.sarah@mayo.edu; 3Department of Oncology, Division of Medical Oncology, Mayo Clinic, Rochester, MN 55902, USA; rmarks@mayo.edu; 4Department of Radiation Oncology, Mayo Clinic, Rochester, MN 55902, USA; olivier.kenneth@mayo.edu; 5Division of General Thoracic Surgery, Mayo Clinic, Rochester, MN 55902, USA; cassivi.stephen@mayo.edu

**Keywords:** thymoma, thymic carcinoma, squamous cell carcinoma, mTAP, BAP1, CDKN2A, immunohistochemistry

## Abstract

**Simple Summary:**

Thymic epithelial tumors, including thymomas and thymic carcinomas are malignant neoplasms that occur in the prevascular (anterior) mediastinum. Thymic carcinomas have a worse outcome than thymomas, and, therefore, management and treatment may differ. However, the morphologic distinction between thymomas and thymic carcinomas can be challenging, as has also been shown in reproducibility studies. We aimed to identify immunohistochemical markers that aid in the distinction between thymomas and thymic carcinomas. We found that a panel of CD117 with a cut-off of 10% tumor cell expression, BAP1, mTAP, and TdT was able to predict 88.9% of thymic carcinomas and 77.8% of thymomas (among the studied types A and B3 thymomas and micronodular thymomas with lymphoid stroma). Only a small subset of thymic carcinomas (11.1%) and thymomas (22.2%) could not be predicted with that model.

**Abstract:**

Background: The morphologic distinction between thymic carcinomas and thymomas, specifically types B3, A, and occasionally micronodular thymomas with lymphoid stroma (MNTLS) can be challenging, as has also been shown in interobserver reproducibility studies. Since thymic carcinomas have a worse prognosis than thymomas, the diagnosis is important for patient management and treatment. This study aimed to identify a panel of immunohistochemical (IHC) markers that aid in the distinction between thymomas and thymic carcinomas in routine practice. Materials and Method: Thymic carcinomas, type A and B3 thymomas, and MNTLS were identified in an institutional database of thymic epithelial tumors (TET) (1963–2021). IHC was performed using antibodies against TdT, Glut-1, CD5, CD117, BAP1, and mTAP. Percent tumor cell staining was recorded (Glut-1, CD5, CD117); loss of expression (BAP1, mTAP) was considered if essentially all tumor cells were negative; TdT was recorded as thymocytes present or absent (including rare thymocytes). Results: 81 specimens included 44 thymomas (25 type A, 11 type B3, 8 MNTLS) and 37 thymic carcinomas (including 24 squamous cell carcinomas). Using BAP1, mTAP, CD117 (cut-off, 10%), and TdT, 88.9% of thymic carcinomas (95.7% of squamous cell carcinomas) and 77.8% of thymomas could be predicted. Glut-1 expression was not found to be useful in that distinction. All tumors that expressed CD5 in ≥50% of tumor cells also expressed CD117 in ≥10% of tumor cells. In four carcinomas with homozygous deletion of *CDKN2A*, mTAP expression was lost in two squamous cell carcinomas and in a subset of tumor cells of an adenocarcinoma and was preserved in a lymphoepithelial carcinoma. Conclusion: A panel of immunostains including BAP1, mTAP, CD117 (using a cut-off of 10% tumor cell expression), and TdT can be useful in the distinction between thymomas and thymic carcinomas, with only a minority of cases being inconclusive.

## 1. Introduction

Thymic epithelial tumors (TET), including thymomas and thymic carcinomas, are uncommon neoplasms that arise from epithelial cells of the thymic gland and occur in the prevascular (anterior) mediastinum. In adults, thymomas represent 31% of all solitary lesions in the prevascular mediastinum [1]. Thymic carcinomas are much less common, accounting for only 7.5% of all solitary prevascular mediastinal lesions in adults [1].

All TET are considered malignant, given their potential to metastasize or recur. Tumors may also cause local complications due to compression of large airways, superior vena cava, and potentially the heart. However, in general, thymic carcinomas behave more aggressively than thymomas [2,3,4]. The estimated 5-year overall survival and recurrence/metastasis-free survival were reported as 77 and 90%, respectively, in thymomas [5] vs. only 36 to 64% and 34 to 41%, respectively, in thymic carcinomas [6,7,8,9,10]. Furthermore, 65 to 79% of patients with thymic carcinoma present at stages III or IV, and complete resection was reported in only 46 to 69% of patients [6,7,8,9,10,11]. In contrast, only 28 to 38% of patients with thymoma present at stages III or IV, and 92% of patients undergo complete resection of thymoma [5,12]. Patients with micronodular thymoma with lymphoid stroma (MNTLS), in general, present at a low tumor stage. Only a rare MNTLS has been reported with pleural implants and wide invasion [13,14]. Recurrences, distant metastases, or tumor-related deaths after resection have not been reported in MNTLS [13,14]. Therefore, the distinction between thymoma and thymic carcinoma is important, as the management and treatment of patients with thymic carcinomas may differ from patients with thymomas [15].

Thymomas are morphologically classified by the World Health Organization (WHO) [16,17]. Both, type A and B3 thymomas lack significant numbers of thymocytes. These two thymoma subtypes are differentiated based on the shape of the tumor cell, which is oval or spindled and bland in type A and polygonal and often with cytologic atypia in type B3 thymoma. MNTLS are characterized by nests (“nodules”) of bland appearing, oval, or slightly spindled epithelial cells. In contrast to other thymoma subtypes, the lymphocytic component of MNTLS is predominantly comprised of B lymphocytes focally forming follicles with germinal centers [17]. Terminal deoxynucleotidyl transferase (TdT)-positive thymocytes are scattered throughout the lymphocytic background. In contrast to the lobulated architecture of thymomas, thymic carcinomas are characterized by a distorted architecture, often with irregular small tumor cell nests, desmoplasia, and increased cytologic atypia [16,17].

Despite the morphologic definitions of these tumors by the WHO, the distinction between some WHO subtypes can be difficult in practice. Indeed, we and others have shown that the interobserver reproducibility between thymic carcinomas and thymomas can be challenging [6,18,19,20,21,22,23]. For instance, in a study of 456 TET reviewed by three thoracic pathologists, we found only a substantial agreement with a κ of 0.65 [18]. Another study of 305 TET evaluated by 13 pathologists with expertise in TET also found only substantial agreement with a similar κ of 0.68 [19]. The distinction between thymoma and thymic carcinoma was noted as especially difficult. Problem areas included the distinction of type B3 thymoma vs. carcinoma, type A vs. B3 thymoma, and occasionally type A thymoma vs. carcinoma and type A thymoma vs. MNTLS. That was further supported by a study in which 95 TET were reviewed by 17 pathologists with expertise in TET and from different institutions also only achieved a moderate agreement with a κ of 0.45 overall and a κ of 0.48 for the distinction of type B3 thymoma vs. thymic carcinomas [22]. This interobserver variation has clinical implications. Indeed, we have shown that, for instance, the interobserver variability resulted in prognostic importance of WHO classification in only 1 (of 3) reviewers [6].

Various immunohistochemical (IHC) studies have attempted to identify a marker or a panel of markers that aid in the distinction of thymoma from thymic carcinoma. Proposed markers included CD5, CD117, Ki-67, beta-5 subunit (beta-5t), and Glucose transporter 1 (GLUT1), among others [20,24,25,26]. While many of these markers are differentially expressed in thymomas and thymic carcinomas, only rare markers have been shown to predict thymoma or thymic carcinoma with certainty. For instance, we have shown that a Ki-67 labeling index of greater than 13.5% or less than 2% is only seen in thymic carcinomas or type A thymomas, respectively [20]. However, a large group of thymomas, including 86% of type A thymomas and 100% of B3 thymomas, and 25% of thymic carcinomas could not be predicted by that model [20].

Recently, loss of expression of BRCA1 associated protein 1 (BAP1) and/or methylthioadenosine phosphorylase (mTAP) has been described in various carcinomas and mesotheliomas but not in reactive mesothelial proliferation [27,28,29,30,31,32,33]. In addition, pathogenic genomic alterations of *BAP1* have been identified in 8.2% of thymic carcinomas [27]. In mesotheliomas, *MTAP* has been shown to be frequently co-deleted with *CDKN2A*; both genes are located close to each other on chromosome 9p21 [33]. Furthermore, we and others have shown that 18 to 29% of thymic carcinomas harbor homozygous deletion of *CDKN2A* [34,35]. Expression of BAP1 and mTAP have not been studied in TET, but the molecular studies suggest that those markers could be useful in the distinction between thymomas and thymic carcinomas.

Overall, while various markers have been found to be differentially expressed in thymomas and thymic carcinomas, cut-offs have not been defined that could help to predict thymoma or thymic carcinoma in practice. We, therefore, aimed to identify IHC markers and, if necessary, their cut-offs of expression that aid in the practical distinction between thymomas, specifically epithelial cell-rich thymomas such as types A and B3 thymoma and MNTLS and thymic carcinomas. Based on the evidence in the literature and commercially available antibodies, we studied a panel of BAP1, CD5, CD117, GLUT1, mTAP, and TdT.

## 2. Material and Methods

### 2.1. Cohort

An institutional database of TET was searched for resection specimens of type A and B3 thymomas, MNTLS, and thymic carcinomas (1963–2021) with available tissue blocks. All tumors were reviewed by a thoracic pathologist (ACR) and classified according to 2021 WHO [17]. Morphologically challenging cases were excluded from this study. All tumors were staged according to the TNM staging (8th AJCC/UICC staging manual) [36]. Medical records were searched for demographics and outcome. A subset of patients was previously reported [34,37]. In some cases, next-generation sequencing (NGS) analysis was performed using FoundationOne^®^ CDx (Foundation Medicine, Inc, Cambridge, MA, USA).

The study was approved by the Mayo Clinic Rochester Institutional Review Board (#10-003525).

### 2.2. Immunohistochemistry

Formalin-fixed paraffin-embedded (FFPE) tissue blocks were cut at 4 microns. Slides were stained with hematoxylin-eosin (H&E) and antibodies against BAP1 (clone C-4, Santa Cruz Biotechnology, Inc, Dallas, TX, USA), CD5 (SP19, Cell Marque, Rocklin, CA, USA), CD117 (YR145, Cell Marque), GLUT1 (polyclonal, Cell Marque), mTAP (2G4, Abnova, Taipei, Taiwan), and TdT (clone SEN28, Leica Biosystems Newcastle Ltd., Newcastle, UK). Percent tumor cell staining was recorded for CD5, CD117, and GLUT1. For GLUT1, an attempt was made to distinguish between membranous and cytoplasmic staining. Loss of nuclear (BAP1) or cytoplasmic (mTAP) expression was considered if essentially all tumor cells were negative with preserved staining in the benign stromal and endothelial cells [38]. If no such positive internal control was present, the stain was not interpreted [39]. TdT-positive thymocytes were recorded as either present or absent. Rare thymocytes were regarded as absent.

### 2.3. Statistical Analysis

Patient characteristics and staining distributions were summarized with frequencies and percentages or medians and interquartile ranges (Q1 [25th], Q3 [75th] percentiles) or ranges. Comparisons between groups (i.e., thymic carcinomas vs. thymomas) were assessed with Chi-square tests, Fisher’s exact tests (categorical variables), or Kruskal–Wallis tests (ordinal or continuous variables), as appropriate. Cut-offs for CD5 and CD117 to distinguish between thymic carcinomas and thymomas were derived ad hoc based on data that showed that none of the thymomas expressed CD5 or CD117 in ≥50% or ≥10% of tumor cells, respectively. Receiver-operating characteristic (ROC) analysis was used to identify a cut-off for GLUT-1 to optimize the sensitivity and specificity, and the area under the curve (AUC) was reported. Recurrence/metastasis-free survival was summarized with the median time-to-event as well as at 5-years post-surgery with the Kaplan–Meier method, along with 95% confidence intervals (CI). All statistical tests were 2-sided. A *p*-value <0.05 was considered statistically significant. All analyses were conducted using SAS version 9.4 (SAS Institute Inc., Cary, NC, USA).

## 3. Results

### 3.1. Patient Demographics, Clinical Characteristics, and Morphologic Findings

Our study was comprised of 81 TET, including 44 thymomas and 37 thymic carcinomas. Among thymomas, there were 25 type A thymomas, 11 type B3 thymomas, and 8 MNTLS. Patient demographics are summarized in Table 1. There was a slight male predominance of patients with thymic carcinoma (54.1%); patients with thymoma were predominantly female (61.4%); this difference was not statistically significant (*p* = 0.17). The median age of patients with thymic carcinoma was 56.5 years; patients with thymoma were in general older, with a median age of 64.9 years (*p* = 0.03). All thymomas and 68.6% of thymic carcinomas were completely resected. Four recurrent TET were also included.

### 3.2. Results of Immunohistochemical Analysis

The results of IHC markers are summarized in Table 2. Loss of expression of BAP1 and mTAP was observed in a subset of thymic carcinomas but not in thymomas. None of the thymic carcinomas showed loss of either marker except for one adenocarcinoma in which tumor cells showed loss of expression of BAP1 and a subset of tumor cells also showed loss of expression of mTAP. Loss of BAP1 occurred in four (of 35, 11.4%) thymic carcinomas, including mucoepidermoid carcinoma, undifferentiated carcinoma, adenocarcinoma (Figure 1A–C), and squamous cell carcinoma (*n* = 1, each). Loss of mTAP expression was observed in five (of 34, 14.7%) thymic carcinomas, including three squamous cell carcinomas (Figure 1D–F), one sarcomatoid carcinoma, and one adenosquamous carcinoma. An additional adenocarcinoma showed loss of mTAP in a subset of tumor cells. This case also showed a loss of expression of BAP1. Expression of GLUT1, CD5, and CD117 was significantly higher in thymic carcinomas (Figure 2A–E) than in thymomas (*p* < 0.001, each). Expression of CD5 in at least 50% of tumor cells and/or CD117 in at least 10% of tumor cells occurred in 10 (of 34, 29.4%) and 28 (of 35, 80%) thymic carcinomas, respectively, and in none of the thymomas. Therefore, a low expression of CD5 and/or CD117 did not exclude the possibility of thymic carcinoma, but high expression supported this diagnosis. All tumors that expressed CD5 in ≥50% of tumor cells also expressed CD117 in ≥10% of tumor cells. GLUT1 expression was not helpful in separating thymic carcinomas from thymomas as no useful cut-off of percent tumor cell expression could be established.
cancers-14-02299-t002_Table 2Table 2Results of immunohistochemistry.Immunohisto-chemical MarkerAssessmentCarcinomaOverallSquamous Cell CarcinomaOther CarcinomaThymoma OverallAB3MNTLS*p*-Value ^a^
*n*3723144425118
BAP1*n* (%) cases with loss of expression4 (11.4) ^b^1 (4.5) ^c^3 (23.1) ^c^0 (0.0) ^d^0 (0.0) ^e^0 (0.0) ^b^0 (0.0) ^c^0.05mTAP5 (14.7) ^f,g^3 (13.6) ^c^2 (16.7) ^b,g^0 (0.0)0 (0.0)0 (0.0)0 (0.0)0.005GLUT1, any stainingMedian % positive tumor cells (range)90 (0–100) ^b^90 (10–100) ^c^40 (0–100) ^c^1 (0–90)5 (0–80)30 (0–90)0 (0–1)<0.0001GLUT1, membranous60 (0–100) ^h^80 (0–100) ^b^5 (0–100) ^b^0 (0–90) ^b^0 (0–5) ^b^10 (0–90)0 (0–0)<0.0001CD55 (0–100) ^f^40 (0–100) ^c^0 (0–10) ^b^0 (0–30)0 (0–30)0 (0–0)0 (0–1)0.0001CD117100 (0–100) ^b^100 (0–100)60 (0–100) ^b^0 (0–1)0 (0–1)0 (0–0)0 (0–0)<0.0001TdT*n* (%) cases with more than rare thymocytes1 (2.8) ^i,c^0 (0.0) ^i^1 (7.7) ^c^34 (77.3) ^k^18 (72.0) ^i^8 (72.7) ^l^8 (100.0)<0.0001^a^*p*-value between carcinoma and all thymomas. Information not available for ^b^ 2; ^c^ 1; ^d^ 8; ^e^ 5; ^f^ 3; ^h^ 4 cases. ^g^ An additional carcinoma showed partial loss of expression of mTAP. ^i^ 2, ^k^ 3, or ^l^ 1 cases with rare thymocytes.
Figure 1(**A**–**C**). Thymic adenocarcinoma. (**A**) The tumor grows in a solid pattern and is comprised of polygonal tumor cells with ample cytoplasm and round nuclei (**B**). (**C**) BAP1 expression is lost in the tumor cells. Note expression of BAP1 is preserved in endothelial cells. (**D**–**F**). Thymic squamous cell carcinoma. (**D**) Large irregular nests of tumor cells are in a desmoplastic and fibrotic stroma. (**E**) The tumor cells are polygonal and have, at least in some areas, a fair amount of cytoplasm. (**F**) mTAP expression is lost in the tumor cells. Note the preserved expression of mTAP in endothelial and stromal cells. Magnification, H&E × 40 (**A**,**D**), × 200 (**B**,**E**), BAP1 × 200 (**C**), mTAP × 200 (**F**).
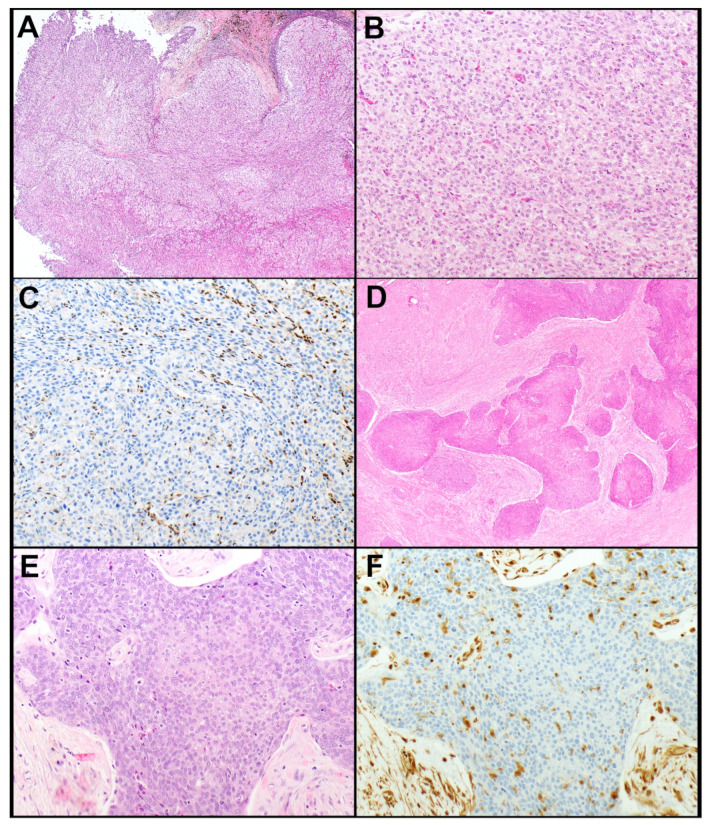

Figure 2Thymic squamous cell carcinoma. (**A**) Irregular nests of tumor cells are growing in a desmoplastic stroma. (**B**) The tumor cells exhibit increased cytologic atypia with slight nuclear pleomorphism, vesicular nuclear chromatin, and conspicuous nucleoli. (**C**) The tumor cells express diffusely GLUT1 (90% of tumor cell expression, membranous; **C**), CD 5 (100% of tumor cell expression; **D**), and CD117 (90% of tumor cell expression; **E**). Magnification, H&E ×40 (**A**), ×200 (**B**), GLUT1 ×200 (**C**), CD5 ×200 (**D**), CD117 ×200 (**E**).
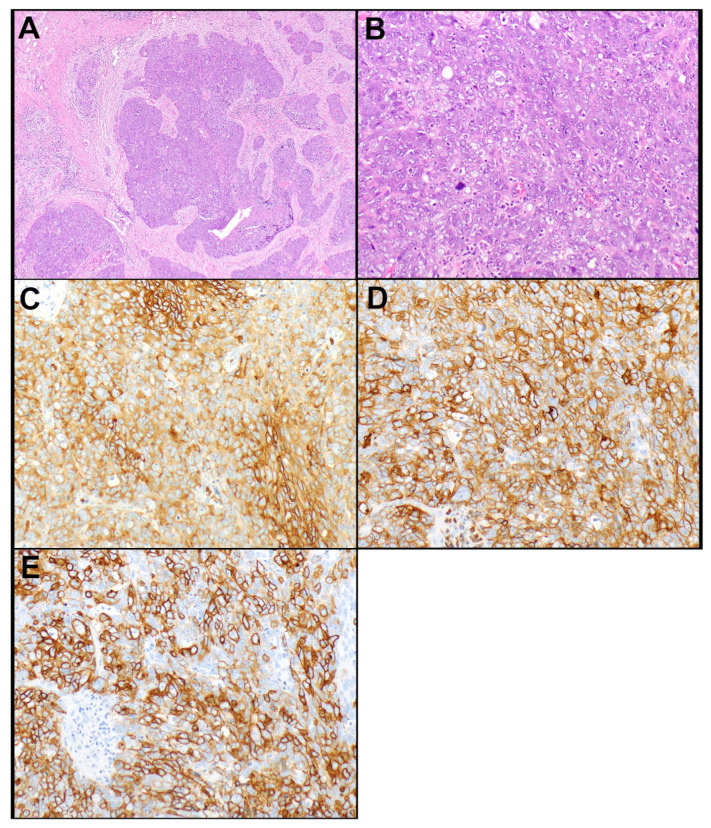


### 3.3. Prediction of Thymic Carcinoma and Thymoma

A large subset of thymic carcinomas (88.9%) could be predicted using a combination of loss of BAP1 and/or mTAP expression and/or expression of CD117 in ≥10% of tumor cells (Figure 3). Among thymic squamous cell carcinomas, the percentage of predicted carcinomas using that panel with the proposed cut-offs were even higher with 22 of 23 (95.7%) tumors being predicted. CD5 was only useful in squamous cell carcinomas in which it was expressed in at least 50% of tumor cells in 10 of 22 (45.5%) tumors. In all other thymic carcinomas in which CD5 was tested, the percent of positive tumor cells was ≤10%. Moreover, CD5 was not found to add any value in the distinction between thymomas and thymic carcinomas as all tumors that expressed CD5 in ≥50% of tumor cells also expressed CD117 in ≥10% of tumor cells. As alluded to earlier, GLUT1 expression was not found to be useful in that distinction. There was no useful cut-off for GLUT1 expression above which only thymic carcinomas could have been diagnosed. ROC curve analysis suggested an optimal cut-off of 40% of tumor cell expression of GLUT1 (AUC = 0.86), which resulted in a sensitivity of 77% and specificity of 80% for the diagnosis of thymic carcinoma. If GLUT1 ≥40% had been added to the proposed algorithm, nine (of 40, 22.5%), thymomas would have been incorrectly predicted as carcinomas. In addition, rare or no TdT-positive thymocytes were not helpful in the prediction of thymic carcinomas as a subset of type A (7 of 25, 28.0%) and B3 thymomas (3 of 11, 27.3%) also lacked clusters of thymocytes. Furthermore, although none of the squamous cell carcinomas showed clusters of thymocytes, a single small cell carcinoma contained clusters of thymocytes (Figure 4A–E). However, this carcinoma was predicted as carcinoma because of the diffuse expression of CD117 (Figure 4F). In addition, morphology and immunophenotype characteristic of a small cell carcinoma was apparent in that case.

Furthermore, 77.8% of thymomas could be predicted as they expressed BAP1 and mTAP, showed no or low (<10%) expression of CD117, and contained clusters of thymocytes, a combination that was not observed in any of the thymic carcinomas. Again, while a single small cell carcinoma did contain clusters of thymocytes, CD117 was expressed in >50% of the tumor cells in that case, therefore, predicting carcinoma. Overall, only 11.1% of thymic carcinomas and 22.2% of thymomas could not be predicted using a panel of BAP1, mTAP, CD117, and TdT (Figure 3).
Figure 3Proposed algorithm using BAP1, CD117, mTAP, and TdT to predict thymic carcinoma and thymomas.
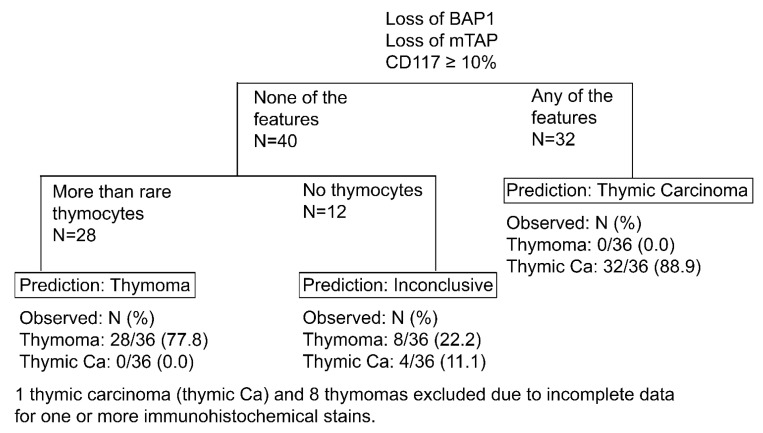

Figure 4Thymic small cell carcinoma. (**A**) This tumor grows in nests and small sheets of tumor cells. (**B**) The tumor cells are medium-sized, have a high nuclear-to-cytoplasmic ratio and dark, finely dispersed nuclear chromatin. (**C**) The tumor cells are negative for GLUT1 (note staining in erythrocytes). (**D**) The tumor cells are also negative for CD5 (note clusters of CD5-positive T lymphocytes/thymocytes). (**E**) TdT highlights clusters of thymocytes in between nests of tumor cells. (**F**) CD117 is expressed in 100% of tumor cells. Magnification, H&E ×40 (**A**), ×200 (**B**), GLUT1 ×200 (**C**), CD5 ×200 (**D**), TdT ×200 (**E**), CD117 ×200 (**F**).
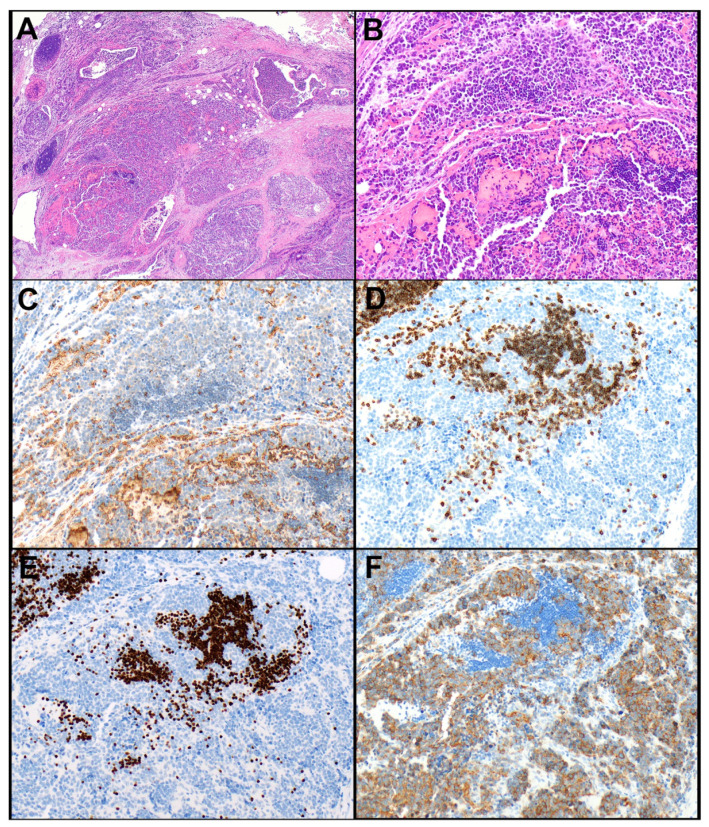


### 3.4. Relationship between Loss of Expression of mTAP and Homozygous Deletion of CDKN2A

In 20 thymic carcinomas, results of our prior study of *CDKN2A* were available for comparison [34]. These 20 thymic carcinomas included squamous cell carcinomas (*n* = 14), lymphoepithelial carcinomas (*n* = 2), adenocarcinomas (*n* = 2), mucoepidermoid carcinoma (*n* = 1), and undifferentiated carcinoma (*n* = 1). Results showed normal signal of *CDKN2A* (*n* = 8), additional copies (*n* = 6), homozygous deletion (*n* = 4), or partial deletion of *CDKN2A* (*n* = 2). In one case with a normal *CDKN2A* signal, mTAP could not be evaluated because of the lack of a positive internal control.

In two cases of thymic carcinoma (both squamous cell carcinomas) with homozygous deletion of *CDKN2A*, mTAP expression was lost; in a lymphoepithelial carcinoma with homozygous deletion of *CDKN2A*, mTAP was expressed. In an adenocarcinoma that was found to have homozygous deletion of *CDKN2A*, mTAP expression was lost in a subset of neoplastic cells (Figure 5A–F). In all other cases in which *CDKN2A* data were available, mTAP was expressed, including both cases with partial deletion of *CDKN2A*.

### 3.5. Comparison of Immunohistochemistry with Next Generation Sequencing

In four cases, including squamous cell carcinoma, small cell carcinoma, mucoepidermoid carcinoma, and type B3 thymoma (one each), NGS testing was available. In the squamous cell carcinoma, NGS identified *CDKN2A/B* loss. This tumor showed preserved expression of mTAP by IHC, and *CDKN2A* FISH revealed normal results. The mucoepidermoid carcinoma harbored a *BAP1* Q40 and a *PPP2R1A* R260C mutation. This tumor showed loss of expression of BAP1 with the preserved expression of mTAP. The small cell carcinoma did not harbor any reportable mutations; BAP1 and mTAP expression were preserved in this tumor. In the B3 thymoma, a *SUZ12* P482fs * 8 mutation was identified.

## 4. Discussion

In this series of 81 TET, including 37 thymic carcinomas and 44 thymomas (including type A and B3 thymomas and MNTLS) we have shown that a panel of CD117, BAP1, mTAP, and TdT can correctly predict 88.9% of thymic carcinomas and 77.8% of thymomas. A cut-off of 10% tumor cell staining for CD117 was found to be useful in predicting thymic carcinoma. Expression of CD5 in 50% or more tumor cells and CD117 in 10% or more tumor cells was only observed in a large subset of thymic carcinomas and not in any thymoma. However, CD5 did not add any value in the distinction between thymic carcinomas and thymomas, as all tumors that expressed CD5 in ≥50% of tumor cells also expressed CD117 in ≥10% of tumor cells. For the prediction of thymic carcinoma, TdT stain was not found to be necessary. Indeed, a single small cell carcinoma did contain clusters of TdT-positive thymocytes; nevertheless, this carcinoma could be correctly predicted because of the diffuse expression of CD117. In contrast to thymic carcinomas, for the prediction of thymomas, clusters of TdT-expressing thymocytes helped in the diagnosis. Only a small percentage of thymomas and thymic carcinomas could not be predicted using this panel of antibodies. Moreover, these four stains are usually validated at least in larger and referral IHC laboratories, and all clones are commercially available. None of our study cases were re-classified based on the results of the IHC stains; this was not surprising as only cases that were morphologically thought to be MNTLS, type A or B3 thymoma, or thymic carcinoma were selected; morphologically challenging cases were excluded.

In contrast to other studies, we did not identify GLUT1, either as membranous and/or cytoplasmic or membranous only expression, as a useful marker to distinguish thymomas from thymic carcinomas in practice. While we confirmed previous reports that the expression of GLUT1 is significantly higher in thymic carcinomas than in thymomas [25], we could not specify a cut-off value that would have allowed us to predict a thymic carcinoma with certainty. A previous study found that GLUT1 may show different expression patterns in thymic carcinomas (strong diffuse) when compared to B3 (moderate to strong, zonal) and type A thymomas (faint, zonal) [40]. However, those patterns were not observed in all such TETs in that study [40]. Therefore, that approach would be difficult to apply in practice.

A recent meta-analysis of 32 IHC markers that may be useful in the distinction of thymomas from thymic carcinomas, and that included 38 articles reporting on 636 thymic carcinomas and 1861 thymomas, showed significant differences in expression between these two entities for beta-5t, Bcl-2, calretinin, CD1a, CD5, CD117, CEA, CK19, GLUT1, IGF-1R, mesothelin, MOC31, MUC1, p21, and TdT [24]. Bcl-2, calretinin, CD5, CD117, CEA, GLUT1, IGF-1R, mesothelin, MOC31, MUC1, and p21 were higher expressed in thymic carcinomas. On the other hand, beta-5t, CD1a, CK19, and TdT showed higher expression in thymomas [24]. In the diagnostic test accuracy review, MUC1 and beta-5t were found to be most useful in distinguishing between thymic carcinomas and thymomas in that meta-analysis [24]. However, that study was not able to show any IHC marker(s) or combination(s) thereof with possible cut-off values that could predict thymic carcinomas or thymomas. Beta-5t appeared to be a promising marker as it was found to be expressed in neoplastic epithelial cells of most type B thymomas, showed variable expression in AB thymomas, and was negative in A thymomas and thymic carcinomas [26]. However, that antibody does not appear to be commercially available and, therefore, is essentially not useful for clinical practice and was not available for our study.

CD5 and CD117 have been previously proposed to aid in the distinction between B3 thymomas and thymic carcinomas [41,42]. However, while more common in thymic carcinomas, these markers were also described to be expressed in B3 thymomas [42,43]. Moreover, many thymic carcinomas are negative for both markers [41,42,44]. We found both markers to be useful in the prediction of thymic carcinomas if CD5 and/or CD117 were expressed in at least 50% or 10% of tumor cells, respectively, even though CD5 did not add any additional value to CD117. We are not aware of any previous studies that proposed a cut-off of expression levels.

Somatic mutations in the *BAP1* gene have been identified in various tumors. For instance, *BAP1* gene mutations have been reported in 23 to 64% of mesotheliomas resulting in nuclear loss of expression of BAP1 protein [28,29]. Nuclear loss of BAP1 protein expression has also been observed in 2 to 50% of carcinomas, including 50% of intrahepatic cholangiocarcinomas [30], 10 to 19% of clear cell renal cell carcinomas [31,32], and 18% of hepatocellular carcinomas [30] among others. A recent study revealed pathogenic genomic alterations in 8.2% of thymic carcinomas, although expression patterns were not studied [27]. In this study, we demonstrated that 11% of thymic carcinomas had lost nuclear expression of BAP1, a finding that we did not observe in any of the tested thymomas. Loss of expression of BAP1 was associated with a *BAP1* mutation in one of the thymic mucoepidermoid carcinomas in which NGS testing was available. Therefore, loss of expression of BAP1 may help to distinguish thymomas from thymic carcinomas even though only a small percentage of thymic carcinomas would be recognized with that marker. Furthermore, preserved expression of BAP1 did not exclude the possibility of thymic carcinoma.

Similarly, a subset of mesotheliomas harbors homozygous deletion of *CDKN2A*, which has not been identified in reactive mesothelial proliferation [28,33]. Specifically, homozygous deletion of *CDKN2A* has been found in 48 to 78% of epithelioid and 67 to 100% of sarcomatoid mesotheliomas [45,46,47]. Interestingly, in 55 to 91% of mesotheliomas, *CDKN2A* deletions have been described in tandem with deletions of *MTAP*, a gene that is located close to *CDKN2A* on chromosome 9p21 [33,46]. In a study of 40 mesotheliomas, homozygous deletion of *CDKN2A* occurred together with homozygous (*n* = 5) or heterozygous (*n* = 5) deletion of *MTAP* or normal *MTAP* (*n* = 3) [33]. Loss of cytoplasmic expression of mTAP was found to be 75% sensitive and 95% specific for heterozygous or homozygous *MTAP* deletion and 59% sensitive and 100% specific for heterozygous or homozygous *CDKN2A* deletion in that study [33]. These findings were supported by another study in mesotheliomas that showed that homozygous deletion of *CDKN2A* was associated with loss of expression of mTAP in all cases studied, resulting in 100% specificity and 74% sensitivity of loss of expression of mTAP for homozygous deletion of *CDKN2A* [48]. However, one study of 17 mesotheliomas identified a single case that showed loss of mTAP expression without an identifiable homozygous *CDKN2A* deletion [46]. The authors suggested that this discordant finding may have been a manifestation of hypermethylation of the *MTAP* gene, as had been reported in other types of tumors. In addition, similar to other studies, two cases were identified with homozygous deletion of *CDKN2A* and preserved mTAP expression in that study [46]. Loss of mTAP expression was also reported in other neoplasms, including 21% of non-small cell lung cancers [49]. We and others have previously shown that 18 to 29% of thymic carcinomas also harbor homozygous deletion of *CDKN2A* [34,35]. Therefore, we evaluated mTAP expression in TET and found a loss of cytoplasmic expression in 15% of thymic carcinomas, most of which (60%) were squamous cell carcinomas. An additional case showed loss of mTAP expression in a distinct subset of the tumor. In cases in which we had results of *CDKN2A* FISH studies and mTAP IHC, we identified a concordant homozygous deletion of *CDKN2A* and loss of expression of mTAP in two squamous cell carcinomas. An additional adenocarcinoma with homozygous deletion of *CDKN2A* showed loss of mTAP expression in a subset of the tumor. This phenomenon is possibly due to various clones or subclones of tumor cells. Similar to studies in mesotheliomas, we also identified one lymphoepithelial carcinoma with the preserved expression of mTAP despite the homozygous deletion of *CDKN2A*. As was suggested in an earlier study, these discordances may occur due to transcriptional or posttranscriptional factors or possible technical issues in the hybridization [46]. However, they may also be due to the deletion of *CDKN2A* with normal *MTAP* gene expression. Interestingly, in one of our thymic squamous cell carcinomas, NGS revealed *CDKN2A/B* loss despite normal *CDKN2A* FISH results and expressed mTAP. This discrepancy may be due to a very small deletion in *CDKN2A*, which cannot be identified by FISH studies and may not lead to co-deletion of *MTAP* or loss of expression of mTAP.

Petrini et al. also identified a homozygous deletion of *CDKN2A* in 10% of B3 thymomas [35]. While we did not perform *CDKN2A* FISH in B3 thymomas, none of the B3 thymomas in our study showed loss of mTAP expression, although the number of cases was relatively small. Overall, our study suggested that loss of cytoplasmic mTAP expression is a predictor of thymic carcinoma even though its sensitivity is low. In addition to being a diagnostic marker, loss of mTAP expression may also be a potential target for therapy, and loss of mTAP expression may help identify candidate patients for such targeted therapy. For instance, in animal studies, it has been shown that the addition of methylthioadenosine to an anti-purine-based chemotherapy can protect mTAP-expressing cells but not mTAP-deficient cells, potentially increasing the therapeutic index of these drugs in tumors that show loss of mTAP expression [50,51]. Furthermore, as mTAP is an essential enzyme for the adenine salvage pathway, it is thought to render tumors susceptible to antifolates targeting de novo purine synthesis. Indeed, a phase II clinical trial treating mTAP-deficient urothelial carcinomas with pemetrexed showed an overall response rate of 43% in addition to a prior cohort in which 4 of 4 mTAP-deficient patients vs. 1 of 10 mTAP-proficient patients responded to pemetrexed [52].

One of the limitations of our study was the long period of time during which specimens were accrued (1963–2021). This was necessary to accumulate enough cases for this study. While all specimens were freshly prepared from FFPE tissue blocks, various fixatives may have been used over the years, and, at least for some immunostains, antigenicity may have decreased over time. In addition, treatment strategies have evolved over the last decades, and, therefore, patients were likely not treated according to the same guidelines, which may have influenced outcome data. Despite these limitations, our study is one of the largest studies to have evaluated IHC markers to aid in the distinction between types A and B3 thymomas and MNTLS versus thymic carcinomas.

## 5. Conclusions

Our study showed that a panel of IHC markers, including BAP1, mTAP, CD117, and TdT can aid in predicting the vast majority of thymic carcinomas and epithelial cell-rich thymomas, including types A and B3 thymomas and MNTLS. These markers may be useful in the practice of pathology when encountering an epithelial cell-rich TET that is difficult to classify on morphology. Larger, ideally multi-institutional studies, are needed to validate our results.

## Figures and Tables

**Figure 5 cancers-14-02299-f005:**
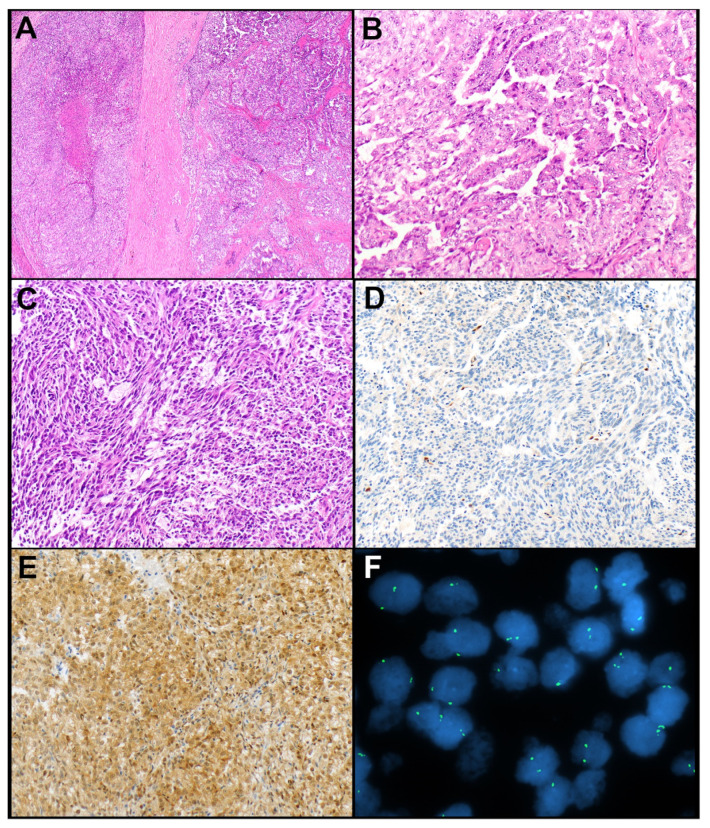
Thymic adenocarcinoma (same case as Figure 1A–C). (**A**) This adenocarcinoma was comprised of solid (left-hand side and Figure 1A,B) and papillary (right-hand side) patterns. (**B**) Neoplastic cells are lining thin fibrovascular cores consistent with a papillary growth pattern. (**C**) In some areas, the tumor was comprised of a spindle cell/sarcomatoid component. (**D**) mTAP expression was lost in the tumor cells of the sarcomatoid component (and the solid component that is shown in Figure 1A,B, respectively, not shown), while mTAP expression was preserved in other areas of the tumor (**E**). (**F**) Homozygous deletion of *CDKN2A* (fluorescence in situ hybridization [FISH], two green signals and no red signal) was observed prepared from the same tumor area that showed loss of expression of mTAP (**D**). FISH studies were not performed in tumor areas that revealed expression of mTAP in tumor cells. Magnification, H&E × 40 (**A**), × 200 (**B**,**C**), mTAP × 200 (**D**,**E**), *CDKN2A* FISH × 1000 (oil) (**F**).

**Table 1 cancers-14-02299-t001:** Demographics, treatment, thymic epithelial tumor subtypes, staging, and outcome of the study population (*n* = 81).

	Carcinoma ^a^	Thymoma ^a^		
		A	B3	MNTLS
Number of patients	37	25	11	8
Male, *n* (%)	20 (54.1)	11 (44.0)	4 (36.4)	2 (25.0)
Age in years, median (Q1, Q3)	56.3	66.4	53.4	74.4
	(48.0, 69.9)	(59.4, 75.3)	(41.4, 65.6)	(65.7, 78.9)
Resection, *n* (%)				
Complete	24 (68.6)	25 (100.0)	9 (100.0)	8 (100.0)
Incomplete	7 (20.0)	0 (0.0)	0 (0.0)	0 (0.0)
Biopsy	4 (11.4)	0 (0.0)	0 (0.0)	0 (0.0)
Thymic carcinoma subtypes, *n* (%)				
Squamous cell carcinoma	24 (64.9)			
Adenocarcinoma	3 (8.1)			
Small cell carcinoma	2 (5.4)			
Undifferentiated carcinoma	2 (5.4)			
Mucoepidermoid carcinoma	2 (5.4)			
Lymphoepithelial carcinoma	2 (5.4)			
Sarcomatoid carcinoma	1 (2.7)			
Adenosquamous carcinoma	1 (2.7)			
Tumor size (if completely resected and primary tumor) in cm, median (Q1, Q3)	5.5 (4.2, 7.5)	5 (3.3, 7.8)	5 (4.2, 7.5)	3.9 (3.0, 4.7)
T-stage, *n* (%)				
1a/b	17 (51.5)	23 (92.0)	7 (77.8)	7 (87.5)
3	16 (48.5)	2 (8.0)	2 (22.2)	1 (12.5)
N-stage, *n* (%)				
0	10 (52.6)	15 (100.0)	7 (100.0)	4 (100.0)
1	8 (42.1)	0 (0.0)	0 (0.0)	0 (0.0)
2	1 (5.3)	0 (0.0)	0 (0.0)	0 (0.0)
M-stage, *n* (%)				
0	27 (79.4)	25 (100.0)	8 (88.9)	8 (100.0)
1a	3 (8.8)	0 (0.0)	0 (0.0)	0 (0.0)
1b	4 (11.8)	0 (0.0)	1 (11.1)	0 (0.0)
TNM stage, *n* (%)				
I	5 (27.8)	15 (100.0)	6 (75.0)	3 (75.0)
IIIA	3 (16.7)	0 (0.0)	1 (12.5)	1 (25.0)
IVA	7 (38.9)	0 (0.0)	0 (0.0)	0 (0.0)
IVB	3 (16.7)	0 (0.0)	1 (12.5)	0 (0.0)
Additional Therapies, *n* (%)				
No	7 (21.2)	23 (92.0)	8 (88.9)	8 (100.0)
Adj radiation	7 (21.2)	1 (4.0)	0 (0.0)	0 (0.0)
Adj chemo	1 (3.0)	0 (0.0)	0 (0.0)	0 (0.0)
Neoadj chemo	1 (3.0)	1 (4.0)	0 (0.0)	0 (0.0)
Neoadj and adj chemo and adj radiation	2 (6.1)	0 (0.0)	1 (11.1)	0 (0.0)
Neoadj and adj radiation	1 (3.0)	0 (0.0)	0 (0.0)	0 (0.0)
Neoadj radiation	2 (6.1)	0 (0.0)	0 (0.0)	0 (0.0)
Neoadj chemoradiation	2 (6.1)	0 (0.0)	0 (0.0)	0 (0.0)
Neoadj and adj radiation and neoadj chemo	1 (3.0)	0 (0.0)	0 (0.0)	0 (0.0)
Adj chemoradiation	7 (21.2)	0 (0.0)	0 (0.0)	0 (0.0)
Neoadj chemo and adj radiation	1 (3.0)	0 (0.0)	0 (0.0)	0 (0.0)
Neoadj and adj chemo and neoadj radiation	1 (3.0)	0 (0.0)	0 (0.0)	0 (0.0)
Follow up				
*n*	35	25	9	8
Median months of follow up (range)	31.5	44.2	57.6	63.2
	(2.2-240.3)	(1.4-238.4)	(1.5-114.0)	(0.1-118.4)
Patients with recurrence/metastasis, *n*	15	0	3	0
Median months to first recur/met	41.2	NA	68.4	NA
5-year recur/met-free survival, % (95% CI) ^b^	39.7	NA	64.3	NA
	(17.7, 61.7)	NA	(23.0, 100.0)	NA
Alive w/o disease, *n*	11	21	5	7
Alive with disease, *n*	5	0	2	0
Died of disease, *n*	7	0	0	0
Months to death of disease, range	2.3-54.5	NA	NA	NA
Died of other cause, *n*	2	2	0	1
Died of unknown cause, *n*	10	2	2	0

^a^ Frequencies not adding to the column total indicate missing data. ^b^ Estimated with the Kaplan–Meier method; Adj, adjuvant; neoadj, neoadjuvant; chemo, chemotherapy; recur, recurrence; met, metastasis; w/o, without; NA, not applicable.

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
