# Peer review of "CD117, BAP1, MTAP, and TdT Is a Useful Immunohistochemical Panel to Distinguish Thymoma from Thymic Carcinoma"

_cancers, 2022, doi:10.3390/cancers14092299_

Round 1

Reviewer 1 Report

The manuscript by Angirekula et al. describes a panel of antibodies to be used in immunohistochemistry to distinguish thymoma from thymic carcinoma.

Overall the manuscript is well written and results are of interest to the pathologist.

A few comments:

1) 2.1 all tumors were reclassified according to the WHO 2021 classification, by one pathologist. What were the criteria used to identify the different groups ? Did these include some immunohistochemical markers ?

2) Based on the results of the IHC, were any of the cases studied, reclassified ?

3) It would be worthwhile mentioning that there are drugs that are being develop to target the MTAP pathway, and that IHC results will be useful to identify gene deletions for patient selection.

4) The number of samples seems to be based on the availability. It would be good to include also a power calculation, allowed by the number of samples collected.

Author Response

The manuscript by Angirekula et al. describes a panel of antibodies to be used in immunohistochemistry to distinguish thymoma from thymic carcinoma. Overall the manuscript is well written and results are of interest to the pathologist.

A few comments:

  • 1 all tumors were reclassified according to the WHO 2021 classification, by one pathologist. What were the criteria used to identify the different groups ? Did these include some immunohistochemical markers ?

Thank you very much for your review of our manuscript, we greatly appreciate it.  The institutional database is based on the WHO classification. As the 2021 WHO classification did not change for types A and B3 thymoma and MNTLS those tumors were easily identifiable based on the subtype of thymoma that is listed in the database. Although the terminology of some thymic carcinomas changed slightly in the 2021 WHO all thymic carcinomas were identifiable based on “carcinoma”. No IHC markers were used to identify these tumors in the database; however such markers may have been used in the process of establishing the primary diagnosis in a few cases.

  • Based on the results of the IHC, were any of the cases studied, reclassified ?

Thank you very much for that question. None of the cases were reclassified based on the results in this study; however, this was a study population with the aim to identify markers that are differentially expressed in type A/B3 thymomas/MNTLS and thymic carcinomas and therefore only included cases that morphologically were not a diagnostic challenge. A sentence has been added to the methods (line 127) and the discussion (lines 327-330).

  • It would be worthwhile mentioning that there are drugs that are being develop to target the MTAP pathway, and that IHC results will be useful to identify gene deletions for patient selection.

Thank you very much, that is an excellent point. We now have included that point in our discussion (lines 421-431).

  • The number of samples seems to be based on the availability. It would be good to include also a power calculation, allowed by the number of samples collected.

Thank you for that suggestion. The number of samples was indeed selected based on availability, and not on a specific power calculation.  Not all type A and B3 thymomas, MNTLS and thymic carcinomas were included in this study as no sufficient tissue was available for some of these cases. Therefore, the number of used cases would not truly reflect the percentage of type A and B3 thymomas, MNTLS and thymic carcinomas in our database. A sentence has been added to the methods (line 125).  Regarding a post-hoc power calculation, as shown in Table 2, most of the stains that we studied differed significantly between the carcinomas and thymomas, therefore, we had sufficient power to detect those differences.  A power calculation for a future study under similar conditions would be appropriate, however, we do not feel that is appropriate here given our findings.

Reviewer 2 Report

The authors of this study set out to establish a panel of immunohistochemistry reactions that could differentiate between thymic carcinoma and thymoma. While the idea is a good one, it may be that the methodology applied is incomplete. It was disappointing to find that factors which are key issues in this type of study were only briefly introduced in the last paragraph of the discussion.

The material for this study was taken from archived paraffin blocks from a period spanning over 50 years. This was unavoidable owing to the scarcity of these tumors, but this introduces room for error which the study has not sufficiently addressed.

Through nearly 60 years, the material in the archive used would have been prepared in different ways. Small changes to fixatives (for example, concentration or buffers), fixation time, and processing procedures, will have made small but potentially significant changes to the tissues.

Antigenicity changes with time, particularly when working with pre-produced sections. The authors underline that the sections were freshly prepared, though they have made little attempt to confirm that the results of their immunohistochemistry are valid using other means. For example, there was mention of the availability of some FISH results, though no mention of extra FISH studies being carried out to confirm deletions or mutations.

BAP1, in particular, is a problematic antigen. Nuclear staining might suggest intact function but the marker may also be found in the cytoplasm (generally associated with poorer prognosis) or staining might be lost altogether (mutation, deletion, artefact). This is a problem that is especially important when working with older samples, particularly those over 20 years old (see Herwig-Carl et al. 2018, for example). The authors of this paper did not assess reactions where staining had failed in tumor cells, but remained in stromal cells. This is a reasonable approach but, given the great age of some of the material, should be backed up by FISH testing to prove the gene has been lost by mutation or deletion and not as a result of an artefact.

Parts of the data presented in the results section are also problematic, given the time period over which the material was collected. The use of survival data is especially questionable. A good deal of the earlier patients will have passed away as a result of any number of causes, not least of which is old age. Indeed, follow-up data were available for 77 of the original 81 cases, and 19 of these had died of either known but unrelated causes or of unknown causes - this is 25% of the group with follow up data or 23% of the entire cohort. Given the size of the whole group, a gap of this size in the data may lead to a considerable skew. The relevance of this data is questionable and the paper may be stronger without it.

The authors appear to have used the results of the immunohistochemistry to determine whether such reactions can support the original diagnoses. It is not clear whether the group attempted any kind of reassessment with pathologists blinded to the existing data. Having formed a hypothesis, albeit a reasonable one, the authors should make certain to show that they have taken suitable steps to rule-out confirmation bias.

MDPI publications, like most others, have rules concerning reference material. MDPI Cancers does not allow authors to self-cite in more than 10% of the reference material. (https://www.mdpi.com/journal/cancers/instructions#references) Two members of this group of authors are cited in more than 10% of the references. This being the case, the authors must review their reference material and either add to, or change the content. If the authors do not feel they can do this, the journal will require a written justification.

Author Response

The authors of this study set out to establish a panel of immunohistochemistry reactions that could differentiate between thymic carcinoma and thymoma. While the idea is a good one, it may be that the methodology applied is incomplete. It was disappointing to find that factors which are key issues in this type of study were only briefly introduced in the last paragraph of the discussion.

The material for this study was taken from archived paraffin blocks from a period spanning over 50 years. This was unavoidable owing to the scarcity of these tumors, but this introduces room for error which the study has not sufficiently addressed.

Through nearly 60 years, the material in the archive used would have been prepared in different ways. Small changes to fixatives (for example, concentration or buffers), fixation time, and processing procedures, will have made small but potentially significant changes to the tissues.

Antigenicity changes with time, particularly when working with pre-produced sections. The authors underline that the sections were freshly prepared, though they have made little attempt to confirm that the results of their immunohistochemistry are valid using other means. For example, there was mention of the availability of some FISH results, though no mention of extra FISH studies being carried out to confirm deletions or mutations.

BAP1, in particular, is a problematic antigen. Nuclear staining might suggest intact function but the marker may also be found in the cytoplasm (generally associated with poorer prognosis) or staining might be lost altogether (mutation, deletion, artefact). This is a problem that is especially important when working with older samples, particularly those over 20 years old (see Herwig-Carl et al. 2018, for example). The authors of this paper did not assess reactions where staining had failed in tumor cells, but remained in stromal cells. This is a reasonable approach but, given the great age of some of the material, should be backed up by FISH testing to prove the gene has been lost by mutation or deletion and not as a result of an artefact.

We would like to thank the reviewer for the thoughtful review. We completely agree that older tissue blocks which may also have been potentially differently fixed may pose a problem in regards to antigenicity. Therefore, as the authors in the manuscript by Herwig-Carl et al point out, we ensured that an internal positive control was present (in addition to positive and negative controls that were performed with the IHC run); any case without such an internal control was not included in the BAP1 results. All loss of BAP1 expression-cases were less than 20 years old, specifically they were 2, 12, and 14 years old. Many other cases that had a similar or older age showed diffuse expression of BAP1. Therefore, overall we feel confident that loss of expression of BAP1 in tumor cells in the presented cases are true results. We now have included the reference of Herwig-Carl et al 2018 into our manuscript.

In addition, we searched medical records of our included cases for NGS results and identified 4 tumors that had undergone NGS. We now present these data in paragraph 3.5. One of the cases harbors a BAP1 mutation; this tumor also showed loss of expression of BAP1. We also added a sentence in the discussion (lines 372-374).

Parts of the data presented in the results section are also problematic, given the time period over which the material was collected. The use of survival data is especially questionable. A good deal of the earlier patients will have passed away as a result of any number of causes, not least of which is old age. Indeed, follow-up data were available for 77 of the original 81 cases, and 19 of these had died of either known but unrelated causes or of unknown causes - this is 25% of the group with follow up data or 23% of the entire cohort. Given the size of the whole group, a gap of this size in the data may lead to a considerable skew. The relevance of this data is questionable and the paper may be stronger without it.

This is a good point and after re-consideration we decided to leave the survival analysis out of this study. Therefore paragraph 3.5 was deleted and the paragraph in the discussion. Paragraph 3.5 was replaced with NGS data.

The authors appear to have used the results of the immunohistochemistry to determine whether such reactions can support the original diagnoses. It is not clear whether the group attempted any kind of reassessment with pathologists blinded to the existing data. Having formed a hypothesis, albeit a reasonable one, the authors should make certain to show that they have taken suitable steps to rule-out confirmation bias.

Thank you for pointing that out. This study was a proof-of-principle-study based on a population of cases in which the diagnosis could be made on morphology alone. We agree, follow-up validation studies are needed to confirm that these stains will be helpful in challenging cases. These validation studies will require a large number of cases as the gold standard for the correct diagnosis will ultimately be outcome. In general, specifically MNTLS, type A thymomas but also many of the type B3 thymomas have a favorable outcome, therefore, a large number of challenging cases will be required for such validation studies which we had tried to convey in the last sentence of the conclusions “Larger, ideally multi-institutional studies are needed to validate our results.”

MDPI publications, like most others, have rules concerning reference material. MDPI Cancers does not allow authors to self-cite in more than 10% of the reference material. (https://www.mdpi.com/journal/cancers/instructions#references) Two members of this group of authors are cited in more than 10% of the references. This being the case, the authors must review their reference material and either add to, or change the content. If the authors do not feel they can do this, the journal will require a written justification.

Thank you for bringing that our attention. We propose 3 references to be removed if felt necessary; unfortunately we can’t supply any references to replace those because of the paucity of studies in this field (please see revised manuscript).

In addition, we identified an oversight in the data of table 1.  As stated, there was 1 patient with stage IVB B3 thymoma. This patient had a metastasis to the heart at time of surgery, therefore, one of the patients with B3 thymoma had M1b disease. This has been updated in table 1.

Round 2

Reviewer 2 Report

The authors have made several changes to the paper which I believe satisfy most points.

A small review of the changes should be carried out as the present form appears to have a few minor errors. In particular lines 157-160 would benefit from review.

Aside from a brief review of typos and spelling, I feel this paper now suitable for further processing.